# BEYOND BROWSING: API-BASED WEB AGENTS

## ABSTRACT

Web browsers are a portal to the internet, where much of human activity is undertaken. Thus, there has been significant research work in AI agents that interact with the internet through web browsing. However, there is also another interface designed specifically for machine interaction with online content: application programming interfaces (APIs). In this paper we ask – *what if we were to take tasks traditionally tackled by browsing agents, and give AI agents access to APIs*? To do so, we propose two varieties of agents: (1) an API-calling agent that attempts to perform online tasks through APIs only, similar to traditional coding agents, and (2) a Hybrid Agent that can interact with online data through both web browsing and APIs. In experiments on WebArena, a widely-used and realistic benchmark for web navigation tasks, we find that API-based agents outperform web browsing agents. Hybrid Agents out-perform both others nearly uniformly across tasks, resulting in a more than 20.0% absolute improvement over web browsing alone, achieving a success rate of 35.8%, achiving the SOTA performance among task-agnostic agents. These results strongly suggest that when APIs are available, they present an attractive alternative to relying on web browsing alone.

## 1 INTRODUCTION

Web agents use browsers as an interface to facilitate humans in performing daily tasks such as online shopping, online planning, trip planning, and other work-related tasks (Liu et al., 2018; Li et al., 2020; Rawles et al., 2023; Patil et al., 2023; Pan et al., 2024; Chen et al., 2024a; Huang et al., 2024; Durante et al., 2024). Existing web agents typically operate within the space of graphical user interfaces (GUI) (Zhang et al., 2023; Zhou et al., 2023; Zheng et al., 2024), using action spaces that simulate human-like keyboard and mouse operations, such as clicking and typing. To observe web pages, common approaches include using accessibility trees, a simplified version of the HTML DOM tree, as the input to text-based models (Zhou et al., 2023; Drouin et al., 2024a), or multi-modal, screenshot-based models (Koh et al., 2024a; Xie et al., 2024; You et al., 2024; Hong et al., 2023). However, regardless of the method of interaction with web sites, there is no getting around the fact that these sites were originally designed for human consumption, and may not be the ideal interface for machines.

Notably, there is another interface designed specifically for machine interaction with online content: application programming interfaces (APIs) (Chan et al., 2024). APIs allow machines to communicate directly with the backend of a web service (Branavan et al., 2009), sending and receiving data in machine-friendly formats such as JSON or XML (Meng et al., 2018; Xu et al., 2021). Nonetheless, whether AI agents can effectively use APIs to tackle real-world online tasks, and the conditions under which this is possible, remain unstudied in the scientific literature. In this work, we explore methods for tackling tasks normally framed as web-navigation tasks with an expanded action space to interact with APIs. To do so, we develop new *API-based agents* that directly interact with web services via API calls, as depicted in Figure 1. This method bypasses the need to interact with web page GUIs through simulated clicks.

At the same time, not all websites have extensive API support, in which case web browsing actions may still be required. To address these cases, we explore a *hybrid* approach that combines API-based agents with web-browsing agents, as described in Figure 1. By implementing an agent capable of *interleaving* API calls and web browsing, we found that agents benefit from the flexibility of this hybrid model. When APIs are available and well-documented, the agent can directly interact with

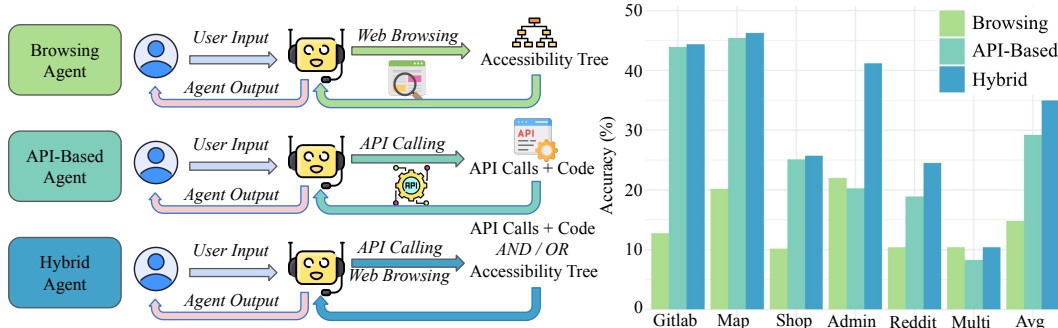

Figure 1: A comparison of three types of agents. The Browsing Agent performs tasks through web browsing only, utilizing the accessibility tree to interact with web pages, achieving an average performance of 14.8% on WebArena. The API-Based Agent performs tasks by making API calls and generating code without relying on web browsing, achieving an average accuracy of 29.2%. The Hybrid Agent combines both methods, dynamically switching between web browsing and API calling, depending on the task. This allows the execution of either API calls or web browsing actions, or both in combination, improving performance by more than 5 percentage points compared to the API-Based Agent .

the web services. For websites with limited API support, the agent seamlessly switches to web browsing mode, simulating human interaction to ensure task completion.

We evaluated our API-based and Hybrid Agents on WebArena, a benchmark for real-world web tasks (Zhou et al., 2023), and the results are shown in Figure 1. Our experiments revealed three key findings: (1) The API-based agent consistently outperforms browsing-based agents on WebArena tasks by around 15% on average, regardless of the comprehensiveness of APIs. (2) The API-based agent yields a higher success rate on websites with extensive API support (e.g., Gitlab) compared to those with limited API support (e.g., Reddit). This result underscores the importance of developing comprehensive API support for more accurate and efficient web task automation in the future. (3) The Hybrid Agent outperforms solely browsing-based agents and solely API-based agents, further improving accuracy by more than 5% compared to the API-based agent. By dynamically switching between approaches, the Hybrid Agent is able to provide more consistent and reliable outcomes.

In sum, our results suggest that allowing agents to interact with APIs, interfaces designed specifically for machines, is often preferable or at least complementary to direct interaction with graphical interfaces designed for humans.

## 2 BACKGROUND: WEB BROWSING

### 2.1 THE WEB BROWSING TASK

Various benchmarks have been developed to evaluate the performance of web browsing agents. MiniWoB (Miniature World of Bits) is an early benchmark that provides simple web-based tasks such as clicking links or typing into forms, but it remains limited in complexity and realism (Shi et al., 2017). Mind2Web scales up these tasks, introducing more sophisticated interactions across websites, but it often lacks the dynamic, real-world scenarios found on the broader web (Deng et al., 2023). WebArena (Zhou et al., 2023) advances web browsing benchmarks by creating reproducible sandboxes of a variety of websites, such as managing repositories, posting online, performing online shopping, and planning trips using map services, while VisualWebArena extends WebArena to the vision modality (Koh et al., 2024a).

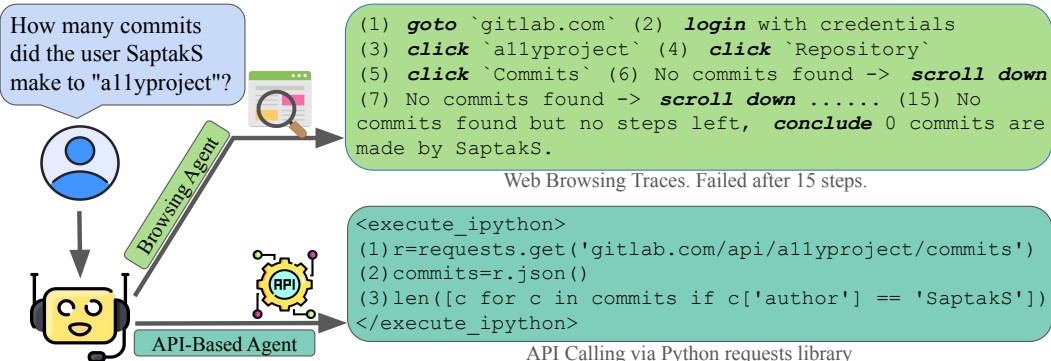

Figure 2: The API-based agent can often solve problems in many fewer function calls than traditional browsing agents . In this task, web browsing failed to solve the intent "find the number of commits the user *SaptakS* made to the repo *a11yproject*" after 15 steps, while our API-based agent successfully completed the task with only three lines of code.

In this paper, we focus on WebArena tasks, which simulate real-world scenarios to evaluate an agent's ability to complete diverse web-based activities.[1] Tasks in WebArena include interacting with platforms like Gitlab (to manage projects and repositories), Reddit (to browse and post content), e-commerce websites (for shopping), and mapping services (for trip planning) (Zhou et al., 2023). Task success is evaluated in three ways: (1) if the task requires producing a specific output, the agent's response is checked for correctness; (2) for tasks involving changes to a website's state (e.g., adding an item to a shopping cart), success is measured by verifying whether the state has changed as expected, such as ensuring the correct item and quantity have been added to the cart; and (3) if the task involves navigation, success is determined by whether the agent reaches the correct URL displaying the desired content.

## 2.2 A BASELINE WEB BROWSING AGENT

While there are a wide variety of agents proposed for such web navigation tasks, in this work we build upon the WebArena baseline agent (Zhou et al., 2023), which operates purely through web interaction by leveraging the accessibility tree[2], a structure that exposes interactive elements like buttons, input fields, and hyperlinks (Yao et al., 2023; Gu et al., 2024). Each element of the accessibility tree is characterized by its functionality such as a hyperlink, its content, and specific web attributes (Liu et al., 2024b; He et al., 2024a; Lù et al., 2024). This exposes web page elements in a hierarchical structure that is easy for agents to navigate (Samuel et al., 2024; Burns et al., 2022).

Agents based on this framework utilize an action space that simulates human browsing behavior, incorporating actions such as simulated clicks, form input, and navigation between pages (Liu et al., 2023; Song et al., 2024; Gur et al., 2024). Importantly, these agents maintain a comprehensive history of their previous actions, allowing them to contextualize their decision-making in past actions.

While agents utilizing this method can navigate arbitrary web pages and often perform well on simpler layouts, challenges arise with the complexity of the accessibility tree. Many large language models (LLMs) are not familiar with this structure, leading to difficulties in completing tasks that require numerous or complex interactions. As a result, the average accuracy hovers in the low double digits (Liu et al., 2024a; Deng et al., 2023; Fu et al., 2024). These methods also struggle with content that need to be dynamically loaded or contents not immediately visible within the tree (Abramovich et al., 2024; Chen et al., 2024b; Lutz et al., 2024).

To give one motivating example, in Figure 2, we demonstrate a task where the agent needs to perform a task determining the number of commits made by the user *SaptakS* in a repository named

---

[1]Notably, upon investigation of VisualWebArena we found that APIs for handling images were relatively limited, and hence we chose to experiment on text-only tasks in this paper.

[2]https://developer.mozilla.org/en-US/docs/Glossary/Accessibility_tree

```
# Commits
## GET /api/{id}/commits: Get a list of commits in a project.
| Attribute | Type           | Description                     |
| `id`      | integer/string | The ID or path of the project.  |
| `since`   | string         | Only commits after or on this date. |
| `until`   | string         | Only commits before or on this date.|
Output: JSON containing all commits that meet the given criteria.
```

```
<execute_ipython>
requests.get('gitlab.com/api/a11yproject/commits')
</execute_ipython>
```

```
[ ......{
    "id": "ed37a2f2",
    "created_at": "2023-03-13T21:04:49.000-04:00",
    "title": "Update README.md",
    "message": "Update README.md",
    "author": "SaptakS",
}]
```

Figure 3: An example of API documentation showing how to get commits of a project, the API call using a Python script to retrieve commits from a project repository, and the resulting JSON response.

*a11yproject*. Specifically, for each task, the agent is given a fixed number of steps within which it has to finish the task. Using a traditional web-browsing approach, the agent follows a complex trajectory, starting with logging into the website, navigating to the correct project, accessing the repository, and finally attempting to view the list of commits. However, due to the large number of commits made by other users, the commits by *SaptakS* are located much further down on the web page, requiring the agent to scroll down many times. As a result, despite completing 15 actions, the browsing agent is unable to retrieve the required information.

## 3 FROM WEB BROWSING TO API CALLING

In contrast to browsing, API calling offers a direct interface for machines to communicate with web services, reducing operational complexity. In this section, we explore an API-based approach when performing web tasks.

### 3.1 APIS AND API DOCUMENTATION

For websites that offer API support, pre-defined endpoints can be utilized to perform tasks efficiently. These APIs, following standardized protocols like REST[3], allow interaction with web services through sending HTTP requests (e.g., GET, POST, PUT) and receiving structured data such as JSON objects[4] as responses. Websites often provide official documentation for the APIs, which can give guidance on how to utilize the APIs. Some documentation is provided in README [5] format, some are in OpenAPI YAML[6] format, and some are in plain text format. For instance, Figure 3 shows the official README documentation of a Gitlab API GET /api/{id}/commits. It documents the functionality, input arguments, and output types of the API. For example, one could use the Python requests library, by calling requests.get("gitlab.com/api/a11yproject/commits"), to retrieve all commits of the repository a11yproject. This would return a JSON list containing all the commits to this repo, as shown in Figure 3.

### 3.2 OBTAINING APIS FOR AGENTS

One important design decision is how to obtain APIs for agents to use. The way agents interact with APIs depends heavily on the availability of APIs and quality of API documentation. In this work, we acquired APIs by manually looking up official API documentation on a website, although this

---

[3]https://en.wikipedia.org/wiki/REST
[4]https://www.json.org/json-en.html
[5]https://en.wikipedia.org/wiki/README
[6]https://yaml.org/

process could potentially be automated in the future. We classify the availability of APIs according to the following three scenarios:

**Sufficient APIs and Documentation**    Many websites provide comprehensive API support and well-documented API documentation in YAML or README format. In this case, simply use the APIs/documentation as-is. Figure 3 depicts an example of API documentation.

**Sufficient APIs, Insufficient Documentation**    There are some challenging situations where APIs exist but good documentation is not officially available. In such cases, additional steps may be required to obtain a list of accessible APIs. In this case, we inspected the frontend or backend code of the website to extract undocumented API calls that can still be utilized by the agent. Then, based on the implementation of APIs of the website, leverage an LLM (GPT-4o[7]) to generate these YAML or README files. By prompting GPT-4o with the relevant implementation details of the APIs (for example, the implementation files of the APIs or example traces of API calls), we generate comprehensive documentation, including input parameters, expected outputs, and example API calls.

**Insufficient APIs**    In the more challenging cases, where only minimal APIs are available, it may be necessary to create new APIs. These custom APIs allow agents to perform tasks that otherwise would require manual web browsing steps. In our case, this was necessary for 1 of 5 web sites in the WebArena benchmark that we utilized, such as creating Reddit APIs discussed in Section 6.2.

### 3.3    USING APIS IN AGENTS

Once we have the APIs and documentation, we then need to provide methods to utilize them in agents. We utilize two different methods based on the size of the API documentation.

**One-Stage Documentation for Small API Sets**    For websites with a smaller number of API endpoints[8], we directly incorporate the full documentation into the prompt provided to the agent. This approach of directly feeding the full documentation worked well for websites with a limited number of API endpoints, as it allowed the agent to have immediate access to all the necessary information without the need for a more complex retrieval mechanism.

**Two-Stage Documentation Retrieval for Large API Sets**    For websites with a larger number of endpoints, providing the full documentation directly within the prompt was impractical due to the size limitations of agent inputs. To address this, we employ a two-stage documentation retrieval process, allowing access to only the relevant information as needed, keeping the initial prompt concise.

In the first stage, the user prompt provide a description of the task, with a list of all available API endpoints along with a very brief description of each API. For example, {`"GET /api/{id}/commits"`:  `"Get a list of commits in a project"`}. This initial summary helps facilitating understanding the scope of all the available APIs while staying within the prompt size constraints.

In the second stage, if the model determines that it needs detailed information about one or more API endpoints, it can use a tool called `get_api_documentation`. This tool has a dictionary that maps each API to its API documentation respectively. The dictionary is obtained using pattern match in Python to retrieve substrings related to each endpoints. `get_api_documentation` is able to search the dictionary and retrieve the full README or YAML documentation for any given endpoint by calling `get_api_documentation` with the endpoint's identifier. This may include the input parameters, output formats, and examples of interacting with the endpoint. For example, to retrieve the documentation for `GET /api/id/commits`, the agent would call `get_api_documentation("GET /api/id/commits")`, and an example returned API documentation is the documentation in Figure 3.

This retrieval method allows the agent to make flexible and informed choices during the execution of tasks. If the agent finds that an API does not meet its needs or if it encounters an error, it can

---

[7]https://openai.com/index/hello-gpt-4o/

[8]Specifically, we use a threshold of 100 APIs, but this could be adjusted depending on the supported language model context size.

easily retrieve the documentation for a different API endpoint by calling the function again. This dynamic approach promotes adaptability and minimizes the risk of incorrect API usage when the number of APIs available is large. The prompt can be found in Appendix A.3.

# 4 HYBRID BROWSING+API CALLING AGENTS

We have proposed API-based methods for handling web tasks, but the question arises: given the benefits of API calling, should we discard web browsing altogether? The most obvious bottleneck is that not all websites offer comprehensive API support. Some platforms offer limited or poorly documented APIs (e.g. there is no API for shopping on Amazon[9]), forcing agents to rely on traditional web browsing methods to complete tasks.

To deal with these situations, we propose a hybrid methods that integrates both browsing-based and API-based approaches, and developed a Hybrid Agent capable of interleaving API calls and web browsing, switching dynamically based on task requirements and the available resources. Specifically, for each task, the agent is given the fixed step budget within which it has to finish the task. In each step, the agent could either (1) communicate with humans in natural language to ask for clarification or confirmation, or 2) generate and executes Python code which could include performing API calling, or 3) performs web browsing actions. The agent could choose freely among these three options, depending on the agent's confidence which method could best tackle the task.

The ideal case is that for websites that offer comprehensive API support, the Hybrid Agent can utilize well-documented endpoints to perform tasks more efficiently than it could through web browsing; for websites with limited API support or poorly documented APIs, the Hybrid Agent could rely more on web browsing to fulfill certain tasks. We later find that enabling an agent to interleave API calling and web browsing boost the agent's performance (see Section 6).

**Prompt Construction** The Hybrid Agent's prompt construction extends upon the API-based agent by incorporating both API and web-browsing documentation. Similar to the API-based agent, the Hybrid Agent is provided with a description of available API calls as discussed in Section 3.3. In addition, the Hybrid Agent receives a detailed specification of the web-browsing actions, which mirrors the information given to the browsing agent described in Section 2.2, including a breakdown of all potential browser interactions. It also maintains a history of all its prior steps such that the agent could make more informed actions. The prompt can be found in Appendix A.4.

# 5 EXPERIMENTAL SETUP

## 5.1 DATASET DESCRIPTION

For our experiments, we utilized the WebArena dataset (Zhou et al., 2023) as the primary evaluation benchmark. WebArena is a comprehensive benchmark designed for real-world web tasks, providing a diverse set of websites simulating various online interactions. WebArena tasks reflect common user activities such as navigating websites, performing administrative tasks, and posting online.

The dataset mainly includes five distinct websites, each containing various intents representing different tasks: **Gitlab**, **Map**, **Shopping**, **Shopping Admin**, **Reddit**, and **Multi-Website Tasks**. We include a more detailed descriptions of the tasks in Appendix A.2. This diverse set of websites and tasks within WebArena allows for a comprehensive evaluation of the agents, testing their ability to handle both API-based interactions and web browsing across varied web settings.

## 5.2 API STATISTICS FOR WEBARENA SITES

In this section, we provide a detailed analysis of the API support of the WebArena websites, categorized into three levels: good, medium, and poor. The availability, functionality, and documentation of APIs, as described in Table 1, play a crucial role in the efficiency and flexibility of our agents.

---

[9]https://www.amazon.com

| Websites | Gitlab | Map | Shopping | Admin | Reddit |
|---|---|---|---|---|---|
| Number of Endpoints | 988 | 53 | 556 | 556 | 31 |
| API/Doc Quality | Good | Good | Fair | Fair | Poor |

Table 1: Number of endpoints, quality of API, and documentation quality for WebArena websites.

### 5.2.1 GOOD API SUPPORT

**Gitlab**  For Gitlab, we leveraged the open Gitlab REST APIs[10], consisting of 988 endpoints. These APIs offer extensive coverage across a wide range of functionalities, including repositories, commits, users, merge requests, and issues. This comprehensive API support allows for effective interaction with most tasks required in WebArena, making it one of the best-supported platforms in terms of API availability.

The majority of Gitlab-related tasks can be handled with the provided APIs, with only a small fraction of tasks, such as retrieving the user's Gitlab feed token, not covered by any existing endpoints. Overall, Gitlab's API structure provides robust support.

**Map**  The Map website offers three sets of APIs, each offering distinct functionalities, with a total of 53 endpoints. Although fewer in number compared to Gitlab and Shopping, these APIs still provide significant coverage for the tasks in WebArena.

The first set of APIs, openly available at Nominatim[11], offers essential endpoints for geographic searches. The second set of APIs, from Project OSRM[12], focuses on routing and navigation functionalities. The third set of APIs, available at OpenStreetMap[13], deals primarily with map database operations. This API is rarely used in WebArena tasks but offers capabilities for interacting with OSM data. Despite the smaller number of endpoints compared to other websites, the APIs available for the Map tasks are mostly well-documented and cover most of the essential WebArena use cases.

### 5.2.2 MEDIUM API SUPPORT

**Shopping and Shopping Admin**  The Shopping and Shopping Admin websites share a common set of APIs from the Adobe Commerce API[14], consisting of 556 endpoints. These APIs provide a reasonable level of support for common shopping tasks such as managing products, categories, and customer accounts. However, some features are absent, such as the ability to add items to a wish list, and thus these tasks must be handled via web browsing. Despite this, the API documentation is fairly detailed and covers most core functionalities, making it a solid, though not exhaustive, solution for handling shopping-related tasks.

### 5.2.3 POOR API SUPPORT

**Reddit**  The Reddit tasks in WebArena are based on a self-hosted limited clone of the Reddit website [15], with limited functionalities as compared to the official site. As a result, all of the available APIs are self-implemented, with a best effort to mimic to official Reddit APIs. With only 31 endpoints, this website offers minimal API support and no API documentation, making it the least API-friendly website in the benchmark.

Many critical functionalities, such as searching for specific posts, are missing, leaving agents to rely heavily on web browsing to complete tasks. The limited API support significantly hampers the efficiency of task execution on Reddit, highlighting the need for a hybrid browsing+API approach.

---

[10]Documentation of all Gitlab APIs could be found at https://docs.gitlab.com/ee/api/rest/.

[11]https://nominatim.org/release-docs/develop/api/Overview/

[12]Openly available at https://project-osrm.org/docs/v5.5.1/api

[13]Publicly available at https://wiki.openstreetmap.org/wiki/API_v0.6

[14]https://developer.adobe.com/commerce/webapi/rest/quick-reference/

[15]https://codeberg.org/Postmill/Postmill

## 5.3 API Implementation Details

In this section, we will discuss how we provided the APIs to the agents when evaluating different web applications inside WebArena, where we follow the methodologies as discussed in Section 3.3.

### 5.3.1 One-Stage Documentation for Small API Sets

For websites with fewer than 100 API endpoints, namely the Map and Reddit websites, we directly incorporated the full documentation into the prompt provided to the agent.

In the case of the Map API, the documentation was sourced directly from the public API documentation provided for the website. The only modification made was the addition of an explanation detailing how to make HTTP requests using the requests library in Python for interacting with the Map API's endpoints. This ensured that the agent could comprehend both the structure of the API and how to implement calls programmatically.

For Reddit, since there was no pre-existing documentation for the APIs, we leveraged GPT-4o[16] itself to generate these README files. By prompting GPT-4o with a file containing all implementations of the API endpoints, we generated a README documentation, including input parameters, expected outputs, and example API calls.

### 5.3.2 Two-Stage Documentation Retrieval for Large API Sets

For websites with more than 100 endpoints, such as GitLab, Shopping, and Shopping Admin, we employ a two-stage documentation retrieval process. For GitLab, we obtained the README documentation from the official GitLab REST API documentation site. For the Shopping and Shopping Admin websites, the documentation was provided in the form of an OpenAPI specification, structured in YAML format.

## 5.4 Evaluation Framework

We employed OpenHands as our primary evaluation framework to facilitate the development and testing of our agents (Wang et al., 2024c). OpenHands is an open-source platform designed for creating and evaluating AI agents that interact with both software and web environments, making it an appropriate infrastructure for our proposed methods. The OpenHands architecture supports a variety of interfaces for agents to interact with. Moreover, this framework allows agents to keep a detailed record of past actions in the prompt, enabling agents to execute actions in a way that is consistent with earlier steps. For coding tasks, it implements an agent based on CodeAct (Wang et al., 2024a) that incorporates a sandboxed bash operating system and Jupyter IPython[17] environments, enabling the execution of Python code. Additionally, it includes a BrowsingAgent browsing agent that focuses solely on web navigation. This agent operates within a Chromium web browser powered by Playwright[18], utilizing a comprehensive set of browser actions defined by BrowserGym (Drouin et al., 2024b). However, while the browsing agent can browse websites, and the CodeActAgent make API calls and execute code, there is not an agent that can natively do both. Given this base, we developed two varieties of agents for API-based solving of web tasks.

**API-Based Agent** First, our API-based agent essentially uses the CodeAct architecture (Wang et al., 2024a). In addition to the basic CodeAct framework, we tailor the agent for API calling by adding specialized instructions and examples that guide its understanding of various API endpoints and their usage. At each step, the agent could utilize all previous actions to make informed selection of actions. The prompt of the API-Based Agent is included in the Appendix A.3.

**Hybrid Browsing/API Calling Agent** In addition to the api-based agent, we developed a Hybrid Agent that integrates Chromium web browsing functionalities powered by Playwright into the existing framework of the API-based agent. This Hybrid Agent is provided the prompt describing both the APIs and the browsing actions, allowing for free transitions between API calling and web

---

[16]https://openai.com/index/hello-gpt-4o/

[17]https://ipython.org

[18]https://playwright.dev/

| Agents | Gitlab | Map | Shopping | Admin | Reddit | Multi | AVG. |
|---|---|---|---|---|---|---|---|
| WebArena Base (Zhou et al., 2023) | 15.0 | 15.6 | 13.9 | 10.4 | 6.6 | 8.3 | 12.3 |
| AutoEval (Pan et al., 2024) | 25.0 | 27.5 | 39.6 | 20.9 | 20.8 | 16.7 | 26.9 |
| AWM (Wang et al., 2024e) | 35.0 | 42.2 | 32.1 | 29.1 | 54.7 | **18.8** | 35.5 |
| SteP (Sodhi et al., 2024)[†] | 32.2 | 31.2 | **50.8** | 23.6 | **57.5** | 10.4 | **36.5** |
| Browsing Agent | 12.8 | 20.2 | 10.2 | 22.0 | 10.4 | 10.4 | 14.8 |
| API-Based Agent | 43.9 | 45.4 | 25.1 | 20.3 | 18.9 | 8.3 | 29.2 |
| Hybrid Agent | **44.4** | **45.9** | 25.7 | **41.2** | 28.3 | 16.7 | 35.8 |

Table 2: Performance of Agents across WebArena Websites. [†]Note that SteP uses prompts inspired specifically by WebArena test set tasks, while other methods are task-agnostic. We achieve the highest performance among the task-agnostic agents.

browsing. At each step, the agent can utilize the current state of the browser, all previous actions taken by the agent, and the results of those actions to determine the next course of action. The prompt of the Hybrid Agent is included in the Appendix A.4.

For the browsing, API-based, and Hybrid Agents, we utilized GPT-4o as the base LLM. However, this could be easily changed to other LLMs.

# 6 RESULTS

## 6.1 MAIN RESULTS

The main results of our evaluation, as summarized in Table 2, demonstrate the performance of three different agents across the websites in the WebArena benchmark.

The API-Based Agent consistently performed well, achieving higher scores in most websites compared to the Browsing agent. This agent's strong performance is attributed to its specialized design for API calling, enabling it to efficiently interact with APIs and complete tasks without reliance on browsing. In contrast, the Browsing Agent, which is designed solely for navigating web interfaces, demonstrated significantly lower performance across most domains. It achieved its best scores on Shopping Admin and Map, but struggled more on the other websites.

| Actions | Gitlab | Map | Shopping | Admin | Reddit | Multi | AVG. |
|---|---|---|---|---|---|---|---|
| Browsing only | 7.8 | 3.7 | 38.5 | 2.2 | 17.0 | 8.3 | 14.3 |
| API only | 21.1 | 4.6 | 7.5 | 1.1 | 0.9 | 10.4 | 8.0 |
| Browsing+API | 71.1 | 91.7 | 54.0 | 96.7 | 82.1 | 81.3 | 77.7 |

Table 3: Percentage of Actions (%) that our Hybrid Agent takes for each type of tasks. Each column sums up to 1.

The Hybrid Agent, integrating both API calling and web browsing, outperformed the other agents on many websites. It's ability to dynamically interleave API calling and browsing proved beneficial. API calling delivers high performance for web tasks with well-supported APIs, while web browsing serves as a backup when API endpoints are unavailable or incomplete. Even if the website provides comprehensive APIs, there might be corner cases where APIs are not supportive. In these cases, relying on web browsing is still needed for tasks that would otherwise fail through API-only interactions. Table 3 documents the percentage of actions of our Hybrid Agent. Across all websites, our Hybrid Agent chooses to do both Browsing and API in the same task at least half of the time.

Table 4 documents the accuracy of the Hybrid Agent across websites when performing different choices of actions. It shows consistently high accuracy when choosing API only and API+browsing.

Overall, the results indicate that the Hybrid Agent is the most effective for handling diverse tasks in WebArena, particularly in environments that require a blend of API and browsing actions. The API-Based Agent excels in tasks that are primarily API-driven, while the Browsing Agent is more suitable for simple navigation tasks but lacks the versatility needed for more complex scenarios.

| Choices of Action | Gitlab | Map | Shopping | Admin | Reddit | Multi | AVG. |
|---|---|---|---|---|---|---|---|
| Browsing only | 7.1(1/14) | 50.0(2/4) | 23.6(17/72) | 50.0(2/4) | 11.1(2/18) | 25.0(1/4) | 21.6(25/116) |
| API only | 47.4(18/38) | 40.0(2/5) | 21.4(3/14) | 50.0(1/2) | 0.0(0/1) | 20.0(1/5) | 38.5(25/65) |
| Browsing+API | 47.7(61/128) | 46.0(46/100) | 27.7(28/101) | 40.9(72/176) | 32.2(28/87) | 15.4(6/39) | 38.2(241/631) |

Table 4: The accuracy (%) of the Hybrid Agent across choices of actions for each website, with the number of correct instances / number of total instances in parentheses.

## 6.2 DOES API QUALITY MATTER?

Yes, API quality does significantly impact the performance of the API-based agent. High quality APIs provide comprehensive and well-documented endpoints that enable agents to interact accurately and efficiently with websites. With comprehensive API support, the API-based agent could tackle more tasks through API calling, while the Hybrid Agent could rely less on browsing; on the other hand, clear and detailed documentation allows agents to utilize the APIs effectively, ensuring that requests are accurate, and minimizing potential errors in task execution. For example, the websites Gitlab and Map with the best API support as mentioned in Section 5.2, demonstrates the highest task completion accuracy by the API-based agent and the Hybrid Agent across all websites.

Conversely, low-quality APIs, characterized by incomplete functionality or ambiguous documentation, can significantly degrade performance. In such cases, the absence of necessary endpoints may prevent the API-based agent from completing tasks, forcing the Hybrid Agent to resort to web browsing. Moreover, poorly documented APIs can result in incorrect parameters and headers being used, further reducing the effectiveness of the agent. This highlights the importance for websites to maintain comprehensive and well-documented API support.

An illustrative example of this is the case of Reddit, where the initial performance of the API-based agent was suboptimal due to limited API availability. As depicted in Table 5, initially, Reddit offered only 18 APIs, lacking the major functionality that common online forums have, such as post voting. Recognizing this lim-

| Number of Endpoints | 18 | 31 |
|---|---|---|
| Accuracy on Reddit | 9.4% | 18.9% |

Table 5: Change in performance of the API-Based Agent on Reddit upon incorporating new APIs.

itation, we manually introduced 13 additional APIs including one API on post voting, with our best effort trying to mimic the official Reddit website. This results in a marked improvement in the API-based agent's performance, underscoring the direct correlation between the availability of high-quality APIs and the average performance of the API-based agent.

Moreover, API quality can also correlate with the performance of browsing agents. This may be because websites with good APIs often have clean, user-friendly interfaces, which benefit machine agents when interacting with the web interface. Good API practices suggest a thoughtful design process that tends to carry over into the overall user interface, allowing the browsing agent to more easily parse and interact with the website. As a result, both API-based and browsing agents are able to function more effectively in environments where high API standards are maintained.

## 7 CONCLUSION AND FUTURE WORK

In this paper, we propose new web agents that use APIs instead of traditionally browsers. We found that API-based agents outperform browsing-based counterparts, especially on websites with sufficient API support. Hence we further propose an agent that is capable of switching between using APIs or browsers and empirically outperforms agents that only uses one of the two interfaces.

For future work, we aim to explore methods for automatically inducing APIs (Wang et al., 2024e). These methods could identify and generate API calls for websites lacking formal API support, further expanding the applicability and efficiency of API-based approaches. By automating the discovery and utilization of APIs, we envision even more robust agents capable of handling diverse web tasks with minimal reliance on manual interaction through browsing.

## 8  LIMITATIONS

**Evaluation Benchmark**   In this paper, we evaluate web agents exclusively on WebArena tasks. While WebArena offers realistic and diverse challenges, the number and variety of tasks may be limited. Other benchmarks, such as Webshop (Yao et al., 2022), MiniWoB (Shi et al., 2017), Mind2Web (Deng et al., 2023), WebVoyager (He et al., 2024b), and VisualWebArena (Koh et al., 2024a), provide alternative evaluation platforms. However, as discussed in Section 2.1, WebArena aligns more closely with real-world scenarios and our use case, while other benchmarks lack support for API calling. For example, VisualWebArena is less applicable to our study because WebArena APIs lack support for interacting with images, a core component of VisualWebArena tasks.

**API Availability**   A key limitation of API-based agents is the inconsistent availability and coverage of APIs across websites. Even platforms with extensive API ecosystems, such as GitLab, may lack support for specific functionalities (e.g., retrieving a user's official username from a displayed name), leading to edge cases where API-based agents are unable to complete tasks due to incomplete API support. However, advancements in techniques like Automatic Web API Mining (AWM) Wang et al. (2024e) could potentially address this limitation by automatically generating APIs for unsupported tasks, reducing reliance on manual API creation.

**Incorporating APIs**   Unlike browsing agents, which can adapt to new websites without manual intervention, the API-based agent requires additional effort to integrate the necessary APIs documentation to the action space of the agent for each website. This manual integration process increases complexity, particularly when the agent must support a wide range of websites, limiting scalability compared to agents that rely solely on web browsing for interactions. However, future advancements in automated API scraping and documentation generation could eliminate this bottleneck, allowing for more scalable and flexible API-based agents.

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

# A APPENDIX

## A.1 RELATED WORK

The development of AI agents that interact with the web and APIs has garnered significant research attention. Web browsers, serving as the primary interface for interacting with online content, have long been a focus for AI research. Web-based agents that can navigate websites, extract information, and perform tasks autonomously have been studied extensively, especially in the context of large language models (LLMs) and agents designed to mimic human behavior online.

**Web Navigation Agents** Much prior work has centered around agents that perform web-based tasks using browsing actions (Yao et al., 2022; Lai et al., 2024; Koh et al., 2024b; Pan et al., 2024). These agents are particularly effective in environments where human-like interaction with a user interface is necessary (Drouin et al., 2024b). Frameworks such as WebArena have further refined the evaluation of such agents by providing complex and realistic web navigation tasks (Zhou et al., 2023). Our work explores the Hybrid Agent that combines web browsing with API interactions. While prior work primarily focuses on browsing-only agents, we examine how Hybrid Agents can enhance performance by integrating structured API calls with web navigation.

**Code Generation Agents and Tool Usage** Another stream of research focuses on agents that interact with online content via application programming interfaces (APIs) (Wang et al., 2024d; Patil et al., 2023; Qin et al., 2023; Yuan et al., 2024; Wang et al., 2024b; Du et al., 2024). In this context, works such as CodeAct have pioneered the development of agents that generate and execute code, including API calls, to perform tasks typically reserved for software engineers (Wang et al., 2024a; Zhang et al., 2024; Tang et al., 2024). These API-based agents are optimized for tasks that involve structured data exchanges, allowing them to perform operations more efficiently than traditional web navigation agents (Shen et al., 2024). On the other hand, our work integrates both browsing and API interactions, demonstrating that Hybrid Agents can outperform API-only agents in tasks requiring web navigation. While existing research shows the efficiency of API-based agents, our Hybrid Agent dynamically switches between APIs and web browsing to optimize task performance.

Additionally, we are the first to explore comparative studies of API v.s. Browsing agents on the same websites. We demonstrate that API-based web agents are often more efficient than browsing

agents when APIs are available, leading to significant improvements in performance. This finding is aligned with previous studies that highlight the advantages of structured interactions through APIs compared to unstructured web browsing interactions.

## A.2 WEBARENA TASKS

WebArena includes the following tasks:

- **Gitlab** – 180 instances: This website simulates tasks related to project management and version control, where agents perform tasks like opening issues, handling merge requests, or creating repositories. Example query: Submit a merge request for a11yproject.com/redesign branch to be merged into markdown-figure-block branch, assign myself as the reviewer.

- **Map** – 109 instances: For this website, tasks are centered around navigation, trip planning and queries about distances, requiring the agent to retrieve and interpret map-based data, similar to using real-world map services like Google map. Example query: Tell me the full address of all international airports that are within a driving distance of 50 km to Carnegie Mellon University.

- **Shopping** – 187 instances: This dataset represents typical e-commerce tasks, such as searching for products, adding items to carts, and processing transactions. Example query: Change the delivery address for my most recent order to 77 Massachusetts Ave, Cambridge, MA.

- **Shopping Admin** – 182 instances: This setting involves managing backend administrative tasks for an online store, like managing product inventories, processing orders, or viewing sales reports. Example query: Tell me the the number of reviews that our store received by far that mention term `"satisfied"`.

- **Reddit** – 106 instances: Tasks here are similar to interactions with the official Reddit, where agents need to post comments, upvote or down-vote posts, or retrieve information from threads. Example query: Tell me the count of comments that have received more downvotes than upvotes for the user who made the latest post on the Showerthoughts forum.

- **Multi-Website Tasks** – 48 instances: These examples involve tasks that span across two websites, requiring the agent to interact with both websites simultaneously, adding complexity to the task. Example query: Create a folder named news in gimmiethat.space repo. Within it, create a file named urls.txt that contains the URLs of the 5 most recent posts from the news related subreddits?

## A.3 API-BASED AGENT PROMPT

> **System Prefix**
>
> You are an AI assistant that performs tasks on the web sites. You should give helpful, detailed, and polite responses to the user's queries.
> You have the ability to call site-specific APIs using Python, or browse the website directly.

**API Prompt**

To call APIs, you can use an interactive Python (Jupyter Notebook) environment, executing code with `<execute_ipython>`.
```
<execute_ipython>
print("Hello World!")
</execute_ipython>
```
This can be used to call the Python requests library, which is already installed for you. Here are some hints about effective API usage:

- It is better to actually view the API response and ensure the relevant information is correctly extracted and utilized before attempting any programmatic parsing.

- Make use of HTTP headers when making API calls, and be careful of the input parameters to each API call.

- Be careful about pagination of the API response, the response might only contain the first few instances, so make sure you look at all instances.

The user will provide you with a list of API calls that you can use.

**System Suffix**

The information provided by the user might be incomplete or ambiguous. For example, if I want to search for `"xyz"`, then `"xyz"` could be the name of a product, a user, or a category on the site. In these cases, you should attempt to evaluate all potential cases that the user might be indicating and be careful about nuances in the user's query. Also, do NOT ask the user for any clarification, they cannot clarify anything and you need to do it yourself.

When you think you successfully finished the task, first respond with `Finish[answer]` where you include *only* your answer to the question `[]` if the user asks for an answer, make sure you should only include the answer to the question but not any additional explanation, details, or commentary unless specifically requested.

After that, when you responded with your answer, you should respond with `<finish></finish>`. Then finally, to exit, you can run
```
<execute_bash>
exit()
</execute_bash>
```
Your responses should be concise. The assistant should attempt fewer things at a time instead of putting too many commands OR too much code in one `execute` block.

Include AT MOST ONE `<execute_ipython>`, `<execute_browse>`, or `<execute_bash>` per response.

IMPORTANT: Execute code using `<execute_ipython>`, `<execute_bash>`, or `<execute_browse>` whenever possible.

Below are some examples:

— START OF EXAMPLE —

Examples

— END OF EXAMPLE —

Now, let's start!

**System Prompt**

```
System Prefix + API Prompt + System Suffix
```

**Initial User Prompt**

Think step by step to perform the following task related to gitlab. Answer the question: ***Example WebArena Intent***
The site URL is `Example Site URL`, use this instead of the normal site URL.
For API calling, use this access token: `Example Access Token`.
My username on this website is `Example Username`.
Below is the list of all APIs you can use and their descriptions:
`Example API Documentation.`
Note: Before actually using a API call, *you should call the `get_api_documentation` function in the `utils` module to get detailed API documentation of the API.* For example, if you want to use the API `GET /api/v4/projects/id/repository/commits`, you should first do:
`<execute_ipython>`
`from utils import get_api_documentation`
`get_api_documentation("GET /api/v4/projects/{id}/repository/commits")`
`</execute_ipython>`
This will provide you with detailed descriptions of the input parameters and example output jsons.

## A.4 HYBRID AGENT PROMPT

**System Prefix**

You are an AI assistant that performs tasks on the web sites. You should give helpful, detailed, and polite responses to the user's queries.
You have the ability to call site-specific APIs using Python, or browse the website directly.
IMPORTANT: In general, you must always first try to use APIs to perform the task; however, you should use web browsing when there is no useful API available for the task.
IMPORTANT: After you tried out using APIs, you must use web browsing to navigate to some URL containing contents that could verify whether the results you obtained by API calling is correct.

**API Prompt**

To call APIs, you can use an interactive Python (Jupyter Notebook) environment, executing code with `<execute_ipython>`.
`<execute_ipython>`
`print("Hello World!")`
`</execute_ipython>`
This can be used to call the Python requests library, which is already installed for you. Here are some hints about effective API usage:

- It is better to actually view the API response and ensure the relevant information is correctly extracted and utilized before attempting any programmatic parsing.

- Make use of HTTP headers when making API calls, and be careful of the input parameters to each API call.

- Be careful about pagination of the API response, the response might only contain the first few instances, so make sure you look at all instances.

The user will provide you with a list of API calls that you can use.

**Browsing Prompt**

You can browse the Internet by putting special browsing commands within `<execute_browse>` and `</execute_browse>` (in Python syntax).

For example to select the option `blue` from the dropdown menu with bid `12`, and click on the submit button with bid `51`:

```
<execute_browse>
select_option("12", "blue")
click("51")
</execute_browse>
```

The following actions are available:

```
def goto(url:  str):
  """Navigate to the specified URL.
  Examples:
    goto('http://www.example.com')
  """

def go_back():
  """Navigate back to the previous page.
  Examples:
    go_back()
  """

def go_forward():
  """Navigate forward to the next page.
  Examples:
    go_forward()
  """

def scroll(delta_x:  float, delta_y:  float):
  """Scroll the page by the specified amount.
  Examples:
    scroll(0, 200)
    scroll(-50.2, -100.5)
  """

def fill(bid:  str, value:  str):
  """Fill the input field with the specified value.
  Examples:
    fill('237', 'example value')
    fill('45', 'multi-line example')
    fill('a12', 'example with "quotes"')
  """

def select_option(bid:  str, options:  str | list[str]):
  """Select an option from a dropdown menu.
  Examples:
    select_option("48", "blue")
    select_option("48", ["red", "green", "blue"])
  """
```

**Browsing Prompt - Continued**

```
def click(bid:  str, button:  Literal["left", "middle", "right"] =
"left", modifiers:  list[typing.Literal["Alt", "Control", "Meta",
"Shift"]] = []):
  """Click on an element with the specified button and modifiers.
  Examples:
    click("51")
    click("b22", button="right")
    click("48", button="middle", modifiers=["Shift"])
  """

def dblclick(bid:  str, button:  Literal["left", "middle", "right"]
= "left", modifiers:  list[typing.Literal["Alt", "Control", "Meta",
"Shift"]] = []):
  """Double-click on an element with the specified button and
modifiers.
  Examples:
    dblclick("12")
    dblclick("ca42", button="right")
    dblclick("178", button="middle", modifiers=["Shift"])
  """

def hover(bid:  str):
  """Hover over an element.
  Examples:
    hover("b8")
  """

def press(bid:  str, key_comb:  str):
  """Press a key combination on an element.
  Examples:
    press("88", "Backspace")
    press("a26", "Control+a")
    press("a61", "Meta+Shift+t")
  """

def focus(bid:  str):
  """Focus on an element.
  Examples:
    focus("b455")
  """

def clear(bid:  str):
  """Clear the input field.
  Examples:
    clear("996")
  """

def drag_and_drop(from_bid:  str, to_bid:  str):
  """Drag and drop an element to another element.
  Examples:
    drag_and_drop("56", "498")
  """

def upload_file(bid:  str, file:  str | list[str]):
  """Upload a file to the specified element.
  Examples:
    upload_file("572", "my_receipt.pdf")
    upload_file("63", ["/home/bob/Documents/image.jpg",
"/home/bob/Documents/file.zip"])
  """
```

**System Suffix**

The information provided by the user might be incomplete or ambiguous. For example, if I want to search for `"xyz"`, then `"xyz"` could be the name of a product, a user, or a category on the site. In these cases, you should attempt to evaluate all potential cases that the user might be indicating and be careful about nuances in the user's query. Also, do NOT ask the user for any clarification, they cannot clarify anything and you need to do it yourself.

When you think you successfully finished the task, first respond with `Finish[answer]` where you include *only* your answer to the question `[]` if the user asks for an answer, make sure you should only include the answer to the question but not any additional explanation, details, or commentary unless specifically requested.

After that, when you responded with your answer, you should respond with `<finish></finish>`. Then finally, to exit, you can run

```
<execute_bash>
exit()
</execute_bash>
```

Your responses should be concise. The assistant should attempt fewer things at a time instead of putting too many commands OR too much code in one `execute` block.

Include AT MOST ONE `<execute_ipython>`, `<execute_browse>`, or `<execute_bash>` per response.

IMPORTANT: Execute code using `<execute_ipython>`, `<execute_bash>`, or `<execute_browse>` whenever possible.

Below are some examples:
— START OF EXAMPLE —
`Examples`
— END OF EXAMPLE —
Now, let's start!

**System Prompt**

```
System Prefix + API Prompt + Browsing Prompt + System Suffix
```

**Initial User Prompt**

Think step by step to perform the following task related to gitlab. Answer the question: `***Example WebArena Intent***`

The site URL is `Example Site URL`, use this instead of the normal site URL.

For API calling, use this access token: `Example Access Token`.

For web browsing, You should start from the URL `Example Start URL`, and this webpage is already logged in and opened for you.

My username on this website is `Example Username`.

Below is the list of all APIs you can use and their descriptions:

`Example API Documentation`.

Note: Before actually using a API call, *you should call the `get_api_documentation` function in the `utils` module to get detailed API documentation of the API.* For example, if you want to use the API `GET /api/v4/projects/id/repository/commits`, you should first do:

```
<execute_ipython>
from utils import get_api_documentation
get_api_documentation("GET /api/v4/projects/{id}/repository/commits")
</execute_ipython>
```

This will provide you with detailed descriptions of the input parameters and example output jsons.

IMPORTANT: In general, you must always first try to use APIs to perform the task; however, you should use web browsing when there is no useful API available for the task. IMPORTANT: After you tried out using APIs, you must use web browsing to navigate to some URL containing contents that could verify whether the results you obtained by API calling is correct.

## A.5 CASE STUDIES

In this section, we analyze two contrasting instances as shown in Figure 4 and Figure 5, where the Hybrid Agent and API-based agent exhibited different levels of performance on WebArena tasks. These case studies highlight the strengths and weaknesses of each agent, demonstrating scenarios where hybrid browsing outperforms API-only or browsing-only approaches, as well as cases where the API-based agent excels over the hybrid method.

```
Task: delete all negative reviews for Sybil running short.

(1) goto `admin.com` (2) login with credentials (3) click
`store` (4) click `products` (5) search `Sybil running short`
(6) iteratively click products on search result and see if it's
the product wanted (7) click review 1 (8) if negative, then
delete (9) click review 2 ......
```
Web browsing has complex traces and lower success rate

```
No API for checking and deleting reviews.
```
API Calling failing due to no useful API available to solve the task

```
(1) GET `/api/products` to retrieve all products (2) get the
product URL from `Sybil running short` product in Python (3)
go to product URL (4)click review 1 (5) if negative, then
delete (6) click review 2 ......
```
Hybrid Agent simplifies task traces

Figure 4: The Hybrid Agent succeeds while the browsing agent and API-based agent both fail

**Case 1** One example where the Hybrid Agent succeeded, while both the API-based and browsing agents failed, involved a task from the Shopping Admin domain. The query was to "delete all negative reviews for Sybil running short," a product listed in the shopping admin interface. In this instance, the API-based agent failed because no relevant API endpoints were available for retrieving or deleting reviews. Similarly, the browsing agent failed, as completing this task purely through web navigation required too many steps, as depicted in Figure 4. This complexity made the task challenging for an agent relying solely on web interactions. However, the Hybrid Agent successfully completed the task by leveraging both API and browsing functionalities. An example trace of the Hybrid Agent shown in Figure 4. This case highlights the Hybrid Agent's ability to efficiently combine API calls with web interactions, allowing it to tackle complex multi-step tasks that would be difficult or impossible for solely browsing or solely API-based agents.

```
Task: tell me the email address of the contributor who has
the most commits to `ai`.

(1) goto `gitlab.com` (2) login with credentials (3) click
`projects` (4) click `ai` (5) click `Repository` (6) click
`Commits` (7) For each contributor, count commit number ......
(15) did not find all commits in 15 steps
```
Web browsing has complex traces and lower success rate

```
requests.get('/api/ai/contributors').json()['email']
```
API Calling successfully completed the task after one API call

```
(1) goto `gitlab.com` (2) login with credentials (3) click
`projects` (4) click `ai` ((4) click `Repository` (5) click
`Commits` (6) For each contributor, count commit number ......
(15) did not find all commits in 15 steps
```
Hybrid Agent fails the task as it falsely seeks help from browsing

Figure 5: Case 2: the API-based agent succeeds while the browsing agent and the Hybrid Agent fails.

**Case 2** Conversely, there are instances where the API-based agent outperforms the Hybrid Agent. One such case occurred in the GitLab website, where the task was to "tell me the email address of the contributor who has the most commits to ai." The API-based agent successfully completed this task by utilizing the GET /api/id/contributors API endpoint to retrieve the contributor with the highest number of commits and their associated email address. On the other hand, the Hybrid Agent attempted to solve the task through browsing but encountered significant challenges. Accessing this information through web browsing required navigating GitLab's interface, locating the correct repository and branch, and identifying the top contributor manually, a task that might be too difficult to perform through web navigation alone. As a result, both the browsing agent and the Hybrid Agent failed to complete the task. This case demonstrates an example where API access provides a more straightforward solution than browsing in contexts requiring structured data retrieval.

## A.6 STEPS AND COSTS

Additionally, we use Table 6 to demonstrate the average steps taken and the average cost for each agent to complete WebArena tasks. The breakdown of steps and cost by website is in the Appendix A.6. Figure 7 demonstrates a scatterplot of the average accuracy of each agent on WebArena over their average steps and average cost.

| Browsing Agent | | API-Based Agent | | Hybrid Agent | |
|---|---|---|---|---|---|
| steps | cost | steps | cost | steps | cost |
| 8.4 | $0.1 | 7.8 | $1.2 | 8.9 | $1.5 |

Table 6: Average number of steps and cost of agents on WebArena tasks

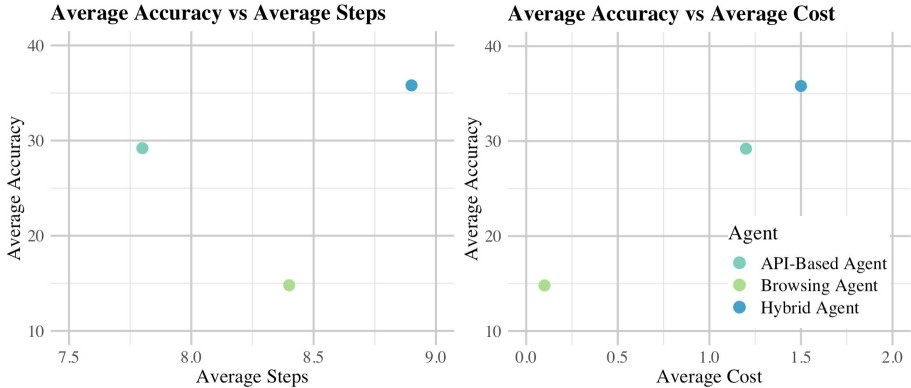

Figure 6: Number of steps (left) and cost (right) of agents averaged across WebArena Websites

**Steps**    The browsing agent takes more steps to complete tasks compared to the API-based agent on average, while the Hybrid Agent takes the most steps amongst the three agents. This is likely due to the browsing agent's reliance on navigating web interfaces and interacting with visual elements, which involves a sequential and more time consuming processes. The API-based agent is the most efficient in terms of steps, as it can directly interact with structured data via APIs, bypassing many of the steps involved in traditional web navigation. The Hybrid Agent, combining both action spaces from the browsing agent and the API-based agent, takes more steps than both agents.

**Costs**    The cost of completing tasks shows a different trend. While the browsing agent requires more steps, it is much cheaper compared to the API-based agent and the Hybrid Agent. This is primarily because the prompts needed for browsing agents are much shorter. When browsing, the agent only needs instructions on how to use the web interface and the limited action space around 14 browsing actions. In contrast, API-based and Hybrid Agents require access to a much larger set of API calls. For example, when interacting with GitLab, the agent is provided with 988 available APIs, leading to much longer prompts and significantly increasing the cost of execution. The cost goes down when the prompt for API calling is shorter. For example, the Reddit website has the least length of API documentation, where its cost is also less than other websites. However, as visualized in Figure 7, the accuracy of the API-based agent and the Hybrid Agent is much higher than the browsing agent, which makes the increase in cost justifiable due to the significantly improved task performance. The higher cost is offset by the agents' ability to complete tasks more accurately and efficiently. In the future, this increased cost could potentially be mitigated by methods that retrieve only relevant APIs on the fly.

Table 7 shows the breakdown of number of steps and cost by website.

| Agents | Gitlab | | Map | | Shopping | | Shop-Admin | | Reddit | | Multi Sites | | AVG. | |
|---|---|---|---|---|---|---|---|---|---|---|---|---|---|---|
| | steps | cost | steps | cost | steps | cost | steps | cost | steps | cost | steps | cost | steps | cost |
| Browsing | 9.4 | 0.2 | 8.0 | 0.1 | 7.3 | 0.1 | 7.0 | 0.2 | 11.1 | 0.1 | 7.5 | 0.1 | 8.4 | 0.1 |
| API-Based | 7.0 | 1.7 | 6.6 | 1.1 | 8.2 | 1.0 | 8.4 | 1.1 | 8.8 | 0.6 | 7.7 | 1.6 | 7.8 | 1.2 |
| Hybrid | 8.1 | 2.0 | 9.4 | 1.7 | 8.2 | 1.3 | 9.0 | 1.4 | 10.5 | 1.0 | 8.0 | 1.9 | 8.9 | 1.5 |

Table 7: Number of Steps and Cost (in U.S. dollars) of Agents across WebArena Websites

## A.7    ERROR ANALYSIS

We randomly sampled 100 tasks from the WebArena tasks and performed error analysis on the API-based agent. We found that 33% of the tasks are correctly performed with only API calling, 50% are unsolvable with solely APIs, 6% are incorrect due to incorrect task understanding, and 11% are incorrect due to error in calling APIs such as mal-formatting and wrong input. In other words, among the 50 API solvable tasks, 66% are performed correctly by the API-based agent. This showcases the strong capability of the API-based agent when given sufficient API to solve the task.

Additionally, the average API calls required to solve the API solvable tasks are 2.1 API calls, demonstrating how API calling could reduce operational complexity for web tasks.

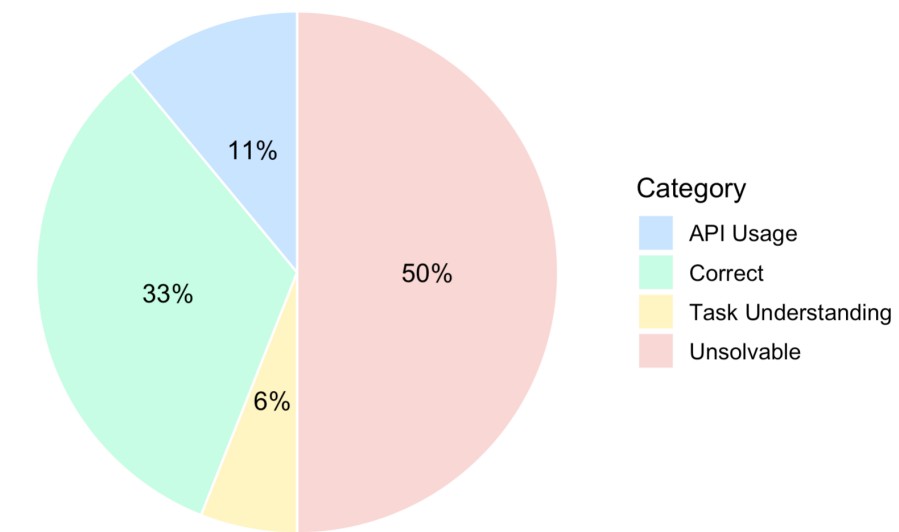

Figure 7: Error analysis on 100 randomly sampled WebArena tasks.

