# OpenReview forum: "Beyond Browsing: API-Based Web Agents"
_ICLR.cc/2025/Conference — ICLR 2025 Conference Withdrawn Submission_

### Official Review · Reviewer_os8n · 2024-11-03

**Soundness:** 2
**Presentation:** 2
**Contribution:** 2
**Rating:** 3
**Confidence:** 4

**Summary:**

This paper proposes a study on leveraging web APIs to improve the performance of web browsing agents. It compares the performance of three settings, web-browsing only, API only, and hybrid ones on WebArena, a recent benchmark for web-based tasks.

**Strengths:**

This paper explores the use of web APIs for web-browsing tasks and studies the performance and cost of different settings.
This paper utilizes the current LLM tools such as ChatGPT, CodeAct, to minimize human interference.

**Weaknesses:**

The major issue of this paper is that it lacks research depth and novelty. Web APIs have been widely used to facilitate the access to online resources, especially data related resources. This paper follows the idea and makes a simple, straightforward attempt of using web APIs if they are available to collect information from websites. No research challenges are identified and the methods are not optimized.

The approach may work in a small-scoped and closed system with the assumption of prior knowledge of the API endpoints and their documentation formats. It is work-in-progress and yet to explore and address many practical problems of leveraging web APIs in an open-end environment. Those problems could be caused due to the lack of systematic supports on locating API endpoints, understanding the semantics of API description, decomposing a task into the invocations of multiple APIs, fixing potential quality issues of codes generated by CodeActAgent, fusing results from multiple APIs, etc.

**Questions:**

1. Invoking web APIs to collect data is a standard use of web APIs. Given this, what is the novelty of the paper? What are the research challenges of including the use of web APIs in a web browsing agent.
2. The proposed two phase documentation retrieval seems to be straightforward and lacks intelligence on interacting with API endpoints. It is unclear how to provide the list of available API endpoints given the prompt of a task? What if there is no readme document or the document is not complete or accurate? Is the agent only designed for RESTful APIs? How about SOAP-based and GraphQL APIs?
3. How to deal with practical issues of using web APIs listed in the weaknesses section?
4. The paper deals with brief and incomplete API documentation by using GPT-4 to generate it. But how reliable the GPT-4  generation would be and what the benefits of using them are unclear.
5. The paper should discuss more about the limitations of the work, in which scenarios human interference is unavoidable, and how to minimize it.
6. The paper compares the proposed approach with StepP, AutoEval, AWM. The paper should describe each work a bit in terms of the methods they propose and why they are selected. The reference items of AutoEval and AWM do not have the journal/conference information so it is unclear whether/where they are published. The paper should also explain why the browsing agent works much less effectively than these three baselines.

---

> ### Author Response · Authors · 2024-11-27
>
> Thank you for your insightful feedback! We would like to address the following concerns:
>
> > Web APIs have been widely used to facilitate the access to online resources, especially data related resources.
>
> It is important to note that although APIs have been widely used for accessing data related resources, it remains underexplored how agents could utilize both APIs and web browsing to perform *arbitrary* real-world web tasks beyond web data scraping.
>
> > This paper follows the idea and makes a simple, straightforward attempt of using web APIs if they are available to collect information from websites. No research challenges are identified and the methods are not optimized.
>
> While the approach may appear simple, this work is the *first* to empirically demonstrate the effectiveness of hybridizing API calling and GUI browsing for real-world web-based tasks. In fact, tool use agents that call APIs and web agents that navigate GUI are studied in two *separated* streams with completely different benchmarks. In this work, we perform an apple-to-apple comparison between these two forms of agents.
> We carefully study the trade-off between API callings and GUI browsing, analysis and discuss the real-world challenges where APIs may not be comprehensive for every website. Our work can lay a foundation for future work to address more complex challenges and call for attention on more machine-friendly interfaces.
>
> > The approach may work in a small-scoped and closed system with the assumption of prior knowledge of the API endpoints and their documentation formats.
>
> We agree that solely API-based agent can be limited, however, our best result comes from the hybrid agent which can flexibly switching between API callings and GUI browsing.uture research could also explore methods for automatic API discovery and documentation generation to extend the approach to more open and dynamic systems.
>
> > The proposed two phase documentation retrieval seems to be straightforward and lacks intelligence on interacting with API endpoints. It is unclear how to provide the list of available API endpoints given the prompt of a task?
>
> The proposed API-based agent is designed as a basic framework, serving as a starting point for future improvements. For instance, future work could include a retrieval step to identify the most relevant API endpoints and provide only their documentation to the agent. Additionally, we have included the prompts we used for the agent in the appendix to offer more clarity.
>
> > What if there is no readme document or the document is not complete or accurate? Is the agent only designed for RESTful APIs? How about SOAP-based and GraphQL APIs?
>
> Thank you for your insightful question! We believe that enabling agents to automatically obtain APIs and adapt to changes of APIs would be an important future work to further support API-based web agents. The agent is not only designed for RESTful APIs. It could also support SOAP-based and GraphQL APIs if provided good documentation on how to use these APIs.
>
> > The paper deals with brief and incomplete API documentation by using GPT-4 to generate it. But how reliable the GPT-4 generation would be and what the benefits of using them are unclear.
>
> Thank you for pointing this out! For websites with no good existing API documentations, we resort to GPT-4o to generate documentation based on the API implementations. To ensure reliability, we manually reviewed the generated documentation to verify its accuracy and completeness. We found that most documents generated by GPT-4 are accurate. This approach provides a practical solution for bridging gaps in documentation quality and availability.
>
> > The paper compares the proposed approach with StepP, AutoEval, AWM. The paper should describe each work a bit in terms of the methods they propose and why they are selected. The reference items of AutoEval and AWM do not have the journal/conference information so it is unclear whether/where they are published. The paper should also explain why the browsing agent works much less effectively than these three baselines.
>
> Thank you for pointing this out! We selected these works based on the WebArena leaderboard (https://webarena.dev/). Specifically, we included the top fully open source agents (at the time of submission) that provide trajectories to compare with. Additionally, AutoEval is recently published on COLM 2024, while AWM is released on Arxiv. The browsing agent is a baseline implemented in OpenHands. It only use few-shot examples without advanced techniques such as language feedback (as in AutoEval), and cross-episod memory (as in SteP and AWM).

---

### Official Review · Reviewer_Ayxo · 2024-11-04

**Soundness:** 2
**Presentation:** 2
**Contribution:** 2
**Rating:** 3
**Confidence:** 4

**Summary:**

This paper proposes to use API-based agents to complement web-only browser based agents for completing web tasks. Using web-arena benchmark which is a simulated web environment, they evaluate how an API-calling agent would perform with focus on highlighting how it has complementary strengths and can outperform a baseline web browsing agent. Finally a hybrid agent is proposed that combines the two agents together to hopefully have the best performance of both the agents.

**Strengths:**

The biggest and (unfortunately) only strength of this paper is the novelty in proposing to use web and API-based agents to highlight the need to use API-based alternative when available and not ONLY rely on web browsing agents alone. This is great idea in practice since it allows for realistic use of the new wave of web agents.

**Weaknesses:**

There are three major weaknesses of the paper as mentioned below along with evidence.
1. Incomprehensive literature survey
- API Agents are not new as mentioned in line 51 on page 1; there has been a lot of work on it such as the ToolBench's ToolLlama, AnyTool's hierarchical agent, etc.
- Sec 9 in page 10 claims that there are no other real-world web task benchmarks available but there are several of them: WebVoyager, WebShop, WorkBench, to name a few.


2. Poorly written and shows that it was not proof-read and submitted in haste.
- Line 047, page 1: "Nonetheless, However, ..."
- Line 083, page 2: "that combining" -> "that combines"
- Line 134, page 3: "withe"?
- Line 157, page 3: "Fig 2" -> "Fig 1"
- Line 189, page 4: "see section ??"
- Table 2 caption, page 8: "Each row columns sums up to 1."-> "Each column sums up to 100"

3. The most important one: imperfect representation/description of the results, which doesn't allow us to be confident of the findings.
- Table 1 description in Sec 6.1 line 375 mentions that the API-based Agent achieves higher scores in all websites compared to the Browsing agent, which is NOT true as can be seen in the results of Shop-Admin and Multi Sites.
- Lines 401-4-3, page 8 say that the browsing agent achieved its best scores on Gitlab and Map, which is not true again. While the sentiment is agreed that it performs poor, it is not represented correctly.
- Line 404, page 8: Hybrid agent did NOT outperform other agents in all categories. In fact, the SteP agent outperforms the hybrid agent in Shopping, Reddit and also on the average!
- Line 423, page 8, "Steps" results description of Table 4 states that "browsing agent consistently takes more steps to complete tasks compared to boath the API-based and hybrid agents". This is false as the table shows that Browsing agent takes least steps than the other two agents for Shopping, Shop-Admin and Multi-Sites.
- Lines 478-479, page 8: This is some leakage that needs to be addressed as adding new APIs are going to obviously help the API-based agents. Also, the numbers are claimed to improved from 9.43% to 14.15% for reddit, but none of the tables show a 9.43% for reddit for any agent.
- WebArena is arguable to be a "realistic" benchmark as WebVoyager is a more realistic one where it is not simulated.

**Questions:**

- A baseline web browsing agent is described in Sec 2.2. This is very little information. There is great amount of work on developing SOTA web browsing agent and full technical reports on that. One of those should be baseline or a description of such a baseline agent shouldn't be possible in 2 paragraphs. Can you please provide more information on the baseline agent OR why one of the SOTA ones is NOT used for the baseline results? Some known agents are WebArena agent, STeP agent, Agent-E, AgentOccam, to name a few.

---

> ### Author Response · Authors · 2024-11-27
>
> Thank you for your insightful feedback! We would like to address the following concerns:
>
> > API Agents are not new as mentioned in line 51 on page 1; there has been a lot of work on it such as the ToolBench's ToolLlama, AnyTool's hierarchical agent, etc.
>
> As discussed in Section A.1 Related Works, our work is fundamentally distinct from the above-mentioned studies, as it specifically focuses on API calling for **web agents**, rather than on generic tool-use agents, where API calling is a natural aspect. We would like to emphasize that in the context of web-based scenarios, our work is the first to propose integrating GUI interactions with API calls.
>
>
> > Sec 9 in page 10 claims that there are no other real-world web task benchmarks available but there are several of them: WebVoyager, WebShop, WorkBench, to name a few.
>
> We added more discussion on why we chose WebArena as the evaluation benchmark in the revised paper. Specifically, in Section 2.1 and Limitations, we discussed existing benchmarks such as MiniWob, Mind2Web, WebVoyager, WebShop, WorkBench, and VisualWebArena, and we discussed why we chose WebArena over the other benchmarks. Specifically, WebArena provides real-world web tasks, and WebArena websites have API support, which allows us to perform an apple-to-apple comparison of web browsing and API calling.
>
>
> > Table 1 description in Sec 6.1 line 375 mentions that the API-based Agent achieves higher scores in all websites compared to the Browsing agent, which is NOT true as can be seen in the results of Shop-Admin and Multi Sites. Lines 401-4-3, page 8 say that the browsing agent achieved its best scores on Gitlab and Map, which is not true again.
>
> We were comparing the WebArena base agent with the API-based agent when we made these statements. We have revised these statements to address the potential confusions.
>
> > In fact, the SteP agent outperforms the hybrid agent in Shopping, Reddit and also on the average!
>
> It is important to highlight that SteP relies heavily on human-designed task templates tailored specifically to the WebArena test set, making it non-task-agnostic. In contrast, our API-based agent and hybrid agent designs are general and do not take any test set into consideration. Therefore, we did not compare our performance with SteP on WebArena. An explanation has been added to Table 2 for clarity.
>
>
> > Lines 478-479, page 8: This is some leakage that needs to be addressed as adding new APIs are going to obviously help the API-based agents.
>
> It is important to note that we did not add new APIs aiming to improve the API-based agent’s performance; instead, we added APIs with our best effort trying to mimic the official Reddit website. This is also an effort that highlights the importance for websites to maintain comprehensive and well-documented API support. Additionally, we added a table 6 in the revised version, where we clarify the number of APIs we added to Reddit and the performance prior to and after adding new APIs.
>
> > A baseline web browsing agent is described in Sec 2.2. This is very little information.
>
> Thank you for pointing this out! We have added more discussion on the baseline browsing agent in Sec 2.2.
>
> > Can you please provide more information on the baseline agent OR why one of the SOTA ones is NOT used for the baseline results?
>
> We would like to point out that we used a SOTA agent from OpenHands as the baseline web browsing agent. We also included several other SOTA agents performances in Table 1 for comparison purposes.

---

### Official Review · Reviewer_fLgW · 2024-11-04

**Soundness:** 3
**Presentation:** 2
**Contribution:** 3
**Rating:** 5
**Confidence:** 4

**Summary:**

The paper introduces an enhanced AI web agent that incorporates API interactions as an additional action space alongside traditional GUI-based interactions. Traditional web agents often rely on simulating human-like actions on graphical user interfaces, which can be inefficient due to the complexity of web pages and limitations in accurately understanding UIs. To address these challenges, the authors develop an API-based agent that interacts directly with web services through API calls, bypassing the need for GUI interaction.

Recognizing that API support varies among websites, the authors also propose a hybrid agent capable of seamlessly switching between API calls and web browsing based on the context. This hybrid approach allows the agent to utilize APIs when available and revert to GUI-based interactions when necessary. The agents are evaluated on the WebArena benchmark, leading to three key findings:
1) API-based agents consistently outperform browsing-based agents on web tasks, regardless of the extent of API support.
2) API-based agents achieve higher success rates on websites with comprehensive API support (e.g., GitLab) compared to those with limited support (e.g., Reddit).
3) Hybrid agents outperform both solely API-based and solely browsing-based agents

**Strengths:**

Originality
The paper introduces a novel approach by incorporating API interactions as an additional action space for AI web agents, which traditionally rely on GUI-based interactions like simulated clicks and typing. By proposing API-based agents and a hybrid model that seamlessly switches between API calls and web browsing, the authors creatively combine existing ideas to address the limitations of current web agents. This innovative problem formulation expands the capabilities of AI agents in interacting with web services and tackles an unstudied area in web task automation.

Quality
The authors provide a robust empirical evaluation of their proposed agents using the WebArena benchmark. The experiments are well-designed to assess the performance across websites with varying levels of API support. The key findings are clearly supported by the data, demonstrating that API-based and hybrid agents consistently outperform traditional GUI-based agents. The paper effectively analyzes the results, highlighting the conditions under which each agent excels, and offers insights into the importance of comprehensive API support.

Clarity
The paper is fairly written. It provides a concise background on the limitations of existing web agents and the motivation for incorporating API interactions. The key contributions are explicitly stated, and the progression from problem statement to conclusion is logical and coherent.

Significance
This work is significant as it addresses a critical gap in the field of AI web agents by leveraging APIs, which are inherently designed for machine interaction. The findings have practical implications for the development of more efficient and accurate web agents capable of handling real-world tasks. By demonstrating that API-based and hybrid agents outperform traditional methods, the paper provides valuable insights that could influence future research and the design of web services. The emphasis on the importance of comprehensive API support underscores a strategic direction for both AI development and web infrastructure enhancement.

**Weaknesses:**

My main concern is the insufficient details on the hybrid agent decision Mechanism: The paper does not provide a detailed explanation of how the hybrid agent decides when to switch between API calls and GUI-based interactions. Clarifying the criteria or algorithms used for this decision-making process is crucial for understanding the agent's functionality and for others to replicate or build upon the work. I personally did not understand how it works in practice. In addition, the paper provides limited technical details on the implementation of the API-based and hybrid agents. For instance, information about the architecture, error handling, and integration with existing systems is sparse. Including more implementation specifics would improve the clarity and allow for better assessment of the work's feasibility and scalability.

Evaluation Scope and Generalizability: The experiments are conducted solely on the WebArena benchmark, which may not cover a sufficiently diverse set of websites and tasks to demonstrate the agent's general applicability. Expanding the evaluation to include a wider variety of websites with different levels of API support and varying complexities would provide stronger evidence of the agent's effectiveness and robustness (See WorkArena, WorkArena++, ST-WebAgentBench, WebCanvas).

Dependence on API Availability and Quality: The proposed approach relies heavily on the availability and comprehensiveness of APIs, which can vary widely across websites. The paper does not address how the agent handles incomplete, undocumented, or changing APIs. Discussing strategies to mitigate these issues, such as API discovery or adaptation mechanisms, would enhance the practicality of the approach.

Security and Ethical Considerations: Direct interaction with web service APIs raises potential security and privacy concerns, such as authentication management, rate limiting, and compliance with terms of service. The paper lacks a discussion of these challenges and does not propose solutions to ensure that the agent operates securely and ethically. Addressing these concerns is important for real-world deployment.

Performance Metrics and Statistical Analysis: The evaluation primarily reports success rates without sufficient analysis of other important performance metrics such as execution time, resource utilization, or learning efficiency. Additionally, the paper does not mention whether the results are statistically significant.

Adaptability to Web Changes: Websites frequently update their interfaces and APIs, which can break automation scripts. The paper does not discuss how the agent adapts to such changes over time. Exploring methods for the agent to detect and adjust to updates would improve its long-term usefulness.

**Questions:**

1. How does the hybrid agent decide when to switch between API calls and GUI-based interactions?

2. How does your agent manage situations with incomplete, undocumented, or frequently changing APIs? Discussing strategies for API discovery, error handling, or adaptation would improve the practicality of your approach.

3. What measures are in place to address security and privacy concerns, such as authentication, rate limiting, and compliance with websites' terms of service?

Can you share detailed information about the technical implementation of your agents, including architecture specifics, error-handling mechanisms, and integration with existing systems?

---

> ### Author Response · Authors · 2024-11-27
>
> Thank you for your insightful feedback! We’re glad to hear that you found the work significant and that it effectively addresses a critical gap in the field of AI web agents through the use of APIs.
> We would like to address the following concerns:
>
> > How does the hybrid agent decide when to switch between API calls and GUI-based interactions? The paper does not provide a detailed explanation of how the hybrid agent decides when to switch between API calls and GUI-based interactions.
>
> Thank you for pointing this out! We added a more detailed explanation on how the hybrid agent works in Section 4, and we provided example prompts to the hybrid agent in the appendix. Specifically, the hybrid agent is provided with both a browsing action space and access to APIs for each website. We use simple prompts and offline the decision-making between browsing and API entirely to the agent. Empirically, the agent usually chooses API whenever it is available, and switch to GUI browsing in other cases.  We also want to highlight that this design is effective, achieving 35.8% task success rate on WebArena, outperforming many existing approaches with heavy human efforts.
>
> > Expanding the evaluation to include a wider variety of websites with different levels of API support and varying complexities would provide stronger evidence of the agent's effectiveness and robustness
>
> We have added more discussion on why we chose WebArena as the evaluation benchmark in the revised paper. Specifically, in Section 2.1 and Limitations, we discussed existing benchmarks such as MiniWob, Mind2Web, WebVoyager, and VisualWebArena, and we discussed why we chose WebArena over the other benchmarks. We have also softened our claim to "Other benchmarks, such as Webshop (Yao et al., 2022), MiniWoB (Shi et al., 2017), Mind2Web (Deng et al., 2023), WebVoyager (He et al., 2024b), and VisualWebArena (Koh et al., 2024a), provide alternative evaluation platforms. However, as discussed in Section 2.1, WebArena aligns closely with real-world scenarios and our use case, while other benchmarks lack support for API calling."
>
> > How does your agent manage situations with incomplete, undocumented, or frequently changing APIs? The paper does not address how the agent handles incomplete, undocumented, or changing APIs.
>
> Our work acknowledges and addresses the limitation of API only agents – We propose hybrid agent, as extensively introduced in Section 4. The corresponding experiments are in Section 5. In such cases, the agent could use web browsing to handle tasks with incomplete, undocumented, or changing APIs, and this hybrid agent achieves the best performance compare to browsing-only and API-only agents.
>
> > Direct interaction with web service APIs raises potential security and privacy concerns, such as authentication management, rate limiting, and compliance with terms of service.
>
> Thank you for pointing this out! We acknowledge the potential risks. We would like to kindly note that addressing the security and privacy concerns are orthogonal to our work.
>
> > The evaluation primarily reports success rates without sufficient analysis of other important performance metrics such as execution time, resource utilization, or learning efficiency.
>
> In Section 6.1, we discussed the average steps and costs the agents take on WebArena tasks. We believe that the average steps serve as a measurement for execution time while costs serves as a measurement for resource utilized. For example, for the shopping admin website, the average cost of the hybrid agent is $1.4, while the average number of steps is 9 steps.
>
> > Websites frequently update their interfaces and APIs, which can break automation scripts. The paper does not discuss how the agent adapts to such changes over time. Exploring methods for the agent to detect and adjust to updates would improve its long-term usefulness.
>
> Thank you for your suggestion! We agree that interface and API update of the websites could bring potential challenges. In our case, we can always provide the *latest* API docs to assist the agent in understanding the new information.

---

> > ### Comment · Reviewer_fLgW · 2024-11-27
> >
> > Thank you for your responses. I understand that running experiments on frontier models can be quite expensive, so I’m fine with your evaluation being limited to WebArena. However, I still have some concerns about the hybrid approach.
> >
> > I would expect a truly hybrid approach to seamlessly combine navigation, data extraction, and API usage within the same workflow. If I understand correctly, your current implementation only makes this decision at the beginning, which seems more like a classifier determining the workflow upfront rather than a dynamic hybrid approach. Is that correct?

---

> > > ### Author Response · Authors · 2024-11-27
> > >
> > > Thank you for your fast response! The current implementation of the hybrid agent could dynamically interleave API calling and web browsing, and it makes the decision of whether to perform API calling or web browsing at each step. In other words, it could perform both actions in one workflow. For example, in the first case study we included in Appendix A.5, Figure 4, it could first perform API calling, and then use the results from API calling to perform web browsing, and vice versa.

---

> > > > ### Comment · Reviewer_fLgW · 2024-11-27
> > > >
> > > > Thank you for the clarification; I understand now.
> > > >
> > > > I believe this aspect should be discussed more thoroughly, clarified, and emphasized in the paper. Without this, as many of the reviewers noted, the novelty of the paper may appear questionable.
> > > >
> > > > To strengthen the paper, I suggest adding concrete examples that illustrate the approach in detail. Provide a full trajectory of how the method works in practice and dive deeper into understanding and explaining the differences compared to other methods. If you have a unique way of integrating the two approaches and can demonstrate that it improves upon the state-of-the-art, this could significantly enhance the paper’s contribution.

---

### Official Review · Reviewer_Pk6m · 2024-11-04

**Soundness:** 2
**Presentation:** 3
**Contribution:** 3
**Rating:** 3
**Confidence:** 4

**Summary:**

The paper presents a hybrid approach that involves use of web browsing and API calling in web agents. The rationale for this approach is that APIs are generally better suited for consumption for agents than the often noisy and expansive HTML DOM structure of webpages.

The result of the experiment on web-arena indicate having good quality API support enables the agent to make use of the APIs to perform the tasks reasonably well by itself or in combination with web browsing.

**Strengths:**

Overall the notion of API + Web browsing shows lot of promise. The implication of this results also presents interesting questions in terms of development of new websites in the future or existing websites in terms of making themselves more agent friendly in terms of exposing capabilities and APIs for optimal consumption by agents.

**Weaknesses:**

In the limitations the authors point that "In our paper we only evaluate web agents on WebArena tasks. The number and diversity of tasks
might be limited. However, to the best of our knowledge, no other real-world web task benchmarks are available at the moment. The tasks we used are the only ones we could find a bemchmark with.". This is not true. There are multiple web benchmarks which are more real-world such as webvoyager, GAIA (a subset of it involves web tasks) and to a limited extend webshop. This is not to say that WebArena evaluation is not valid, i think it is a perfectly reasonable evaluation, however the statement "no other real-world web task benchmarks are available at the moment" is factually not true.
Also not the spelling error in "bemchmark"

Missing analysis: I would also like to see an analysis of the error modes for API and API+Browsing. How many of the tasks could be performed exclusively using APIs and what are the minial number of API calls required for each. How many of these could the agent perform, and for those that the agent could not, what were the common error modes. The authors does mention good quality APIs influence performance ,however this is pretty obvious and generic. A detailed analysis of the error modes would present valuable insights.

**Questions:**

Elaborate on the error modes.

---

> ### Author Response · Authors · 2024-11-27
>
> Thank you for your insightful feedback! We are very happy to hear your recognition that API + browsing is promising. We believe all your questions are addressable during the rebuttal period:
>
> > There are multiple web benchmarks which are more real-world such as webvoyager, GAIA (a subset of it involves web tasks) and to a limited extend webshop.
>
> Thank you for pointing this out! The main reason to choose WebArena over other benchmarks is that WebArena provides *both* GUI browsing and API support in the *same* environment for the *same* set of tasks. This allows us to perform apple-to-apple comparison between GUI browsing and API. Other existing benchmarks usually only support *one* scenario. For example, While WebVoyager support real-world GUI navigation, it does not provide the corresponding API support.
>
> We have added more discussion on why we chose WebArena as the evaluation benchmark in the revised paper. Specifically, in Section 2.1 and Limitations, we discussed existing benchmarks such as MiniWob, Mind2Web, WebVoyager, and VisualWebArena, and we discussed why we chose WebArena over the other benchmarks. We have also softened our claim to "Other benchmarks, such as Webshop (Yao et al., 2022), MiniWoB (Shi et al., 2017), Mind2Web (Deng et al., 2023), WebVoyager (He et al., 2024b), and VisualWebArena (Koh et al., 2024a), provide alternative evaluation platforms. However, as discussed in Section 2.1, WebArena aligns closely with real-world scenarios and our use case, while other benchmarks lack support for API calling."
>
> > I would also like to see an analysis of the error modes for API and API+Browsing. How many of the tasks could be performed exclusively using APIs and what are the minimal number of API calls required for each. How many of these could the agent perform, and for those that the agent could not, what were the common error modes.
>
> Thank you for your suggestion! We randomly sampled 100 tasks from the WebArena tasks, and we found that 33% of the tasks are correctly performed with API calling, 50% are unsolvable with solely APIs, 6% are incorrect due to incorrect task understanding, and 11% are incorrect due to error in calling APIs (such as mal-formatting, wrong input, etc.). In other words, among the 50 API solvable tasks, 66% are performed correctly. Additionally, the average API calls required to solve the API solvable tasks are 2.1 API calls. We uploaded the samples and the error analysis as supplementary materials.
>
> | **Error Type**       | **Count** |
> |------------------|--------------|
> | **Correct**      | 33 |
> | **Unsolvable**| 50 |
> | **Task Understanding**| 6 |
> | **API Usage**| 11 |

---

### Note · Authors · 2024-12-14

I have read and agree with the venue's withdrawal policy on behalf of myself and my co-authors.